# Lemnalol Modulates the Electrophysiological Characteristics and Calcium Homeostasis of Atrial Myocytes

**DOI:** 10.3390/md17110619

**Published:** 2019-10-30

**Authors:** Buh-Yuan Tai, Zhi-Hong Wen, Pao-Yun Cheng, Hsiang-Yu Yang, Chang-Yih Duh, Ping-Nan Chen, Chih-Hsueng Hsu

**Affiliations:** 1Department of Marine Biotechnology and Resources, National Sun Yat-Sen University, Kaohsiung 80424, Taiwan; tai1337@hotmail.com (B.-Y.T.); wzh@mail.nsysu.edu.tw (Z.-H.W.); yihduh@mail.nsysu.edu.tw (C.-Y.D.); 2Department of Traditional Medicine, Jianan Mental Hospital, Tainan 717, Taiwan; 3Department of Physiology and Biophysics, National Defense Medical Center, Taipei 114, Taiwan; pycheng@mail.ndmctsgh.edu.tw; 4Division of Cardiovascular Surgery, Department of Surgery, Tri-Service General Hospital, National Defense Medical Center, Taipei 114, Taiwan; alfie0314@gmail.com; 5Department of Biomedical Engineering, National Defense Medical Center, Taipei 114, Taiwan; g931310@mail.ndmctsgh.edu.tw; 6Division of Cardiology, Tri-Service General Hospital, National Defense Medical Center, Taipei 114, Taiwan

**Keywords:** Lemnalol, calcium homeostasis, inflammation, sepsis, left atrium cardiomyocyte, ionic current

## Abstract

Sepsis, an inflammatory response to infection provoked by lipopolysaccharide (LPS), is associated with high mortality, as well as ischemic stroke and new-onset atrial arrhythmia. Severe bacterial infections causing sepsis always result in profound physiological changes, including fever, hypotension, arrhythmia, necrosis of tissue, systemic multi-organ dysfunction and finally death. LPS challenge-induced inflammatory responses during sepsis may increase the likelihood of the arrhythmogenesis. Lemnalol is known to possess potent anti-inflammatory effects. This study examined whether Lemnalol (0.1 μM) could modulate the electrophysiological characteristics and calcium homeostasis of atrial myocytes under the influence of LPS (1μg/mL). Under challenge with LPS, Lemnalol-treated LA myocytes, had a longer AP duration at 20%, 50% and 90% repolarization of the amplitude, compared to the LPS-treated cells. LPS-challenged LA myocytes showed increased late sodium current, Na^+^-Ca^2+^ exchanger current, transient outward current, rapid component of delayed rectifier potassium current, tumor necrosis factor-α, NF-κB and increased phosphorylation of ryanodine receptor (RyR), but a lower L-type Ca^2+^ current than the control LA myocytes. Exposure to Lemnalol reversed the LPS-induced effects. The LPS-treated and control groups of LA myocytes, with or without the existence of Lemnalol. showed no apparent alterations in the sodium current amplitude or Cav1.2 expression. The expression of sarcoendoplasmic reticulum calcium transport ATPase (SERCA2) was reduced by LPS treatment, while Lemnalol ameliorated the LPS-induced alterations. The phosphorylation of RyR was enhanced by LPS treatment, while Lemnalol attenuated the LPS-induced alterations. In conclusion, Lemnalol modulates LPS-induced alterations of LA calcium homeostasis and blocks the NF-κB pathways, which may contribute to the attenuation of LPS-induced arrhythmogenesis.

## 1. Introduction

The most prevalent arrhythmia in the clinical is atrial fibrillation (AF) [1]. The occurrence of AF is closely associated with the interaction of cardiac tissues and the inflammatory response, including a variety of immune proteins and the cells they mediate. Inflammation is a biological process by which an organism defends against stimuli from the environment. Severe bacterial infections always result in profound physiological alterations, including high fever, hypotension, arrhythmia, necrosis of tissue, systemic multi-organ failure and finally death. A great deal of solid evidence suggests that sepsis is the most prevalent factor associated with the occurrence of AF [2,3,4]. Macrophages activated by lipopolysaccharides (LPS) secrete many inflammatory mediators [5]. Tumor necrosis factor-α (TNF-α), platelet-activating factor (PAF), interleukin (IL)-1β, IL-6, IL-8, IL-10 and chemokines produce an inflammatory effect upon adjacent cardiac tissues [6,7,8]. Inflammatory mediators and activated immune cells work together to cause clinical symptoms, such as arrhythmogenesis, peripheral vascular dilation, thrombogenesis, endothelial damage and fever [9]. The so-called ‘AF begets AF’ hypothesis clearly states that AF can induce inflammatory processes during cardiomyocyte remodeling, both electrical and structural [10,11], forming a vicious cycle. LPS was shown to cause marked increases in systemic inflammatory responses via the nuclear factor NF–κB signaling pathway and associated biochemical and physiological regulation [12,13]. 

Lemnalol (8-isopropyl-5-methyl-4-methylene-decahydro-1,5-cyclo-naphthalen-3-ol) is a ylangene-type sesquiterpenoid compound originally isolated from the soft coral *Lemnalia tenuis* by Kikuchi et al. [14]. Duh et al. [15] also isolated Lemnalol from *Lemnalia cervicorni*. It shows anti-inflammatory and anti-tumor activities [16,17]. Our previous studies had found that Lemnalol inhibits pro-inflammatory inducible nitric oxide synthase (iNOS) and cyclooxygenase-2 (COX-2) protein expression in LPS-treated macrophages [16]. Moreover, it also produced significant anti-nociceptive effects in carrageenan-induced inflammation, chronic constriction injury-induced neuropathy and monosodium urate-induced gouty arthritis in rats. [18,19]. There are several mechanisms with evidence indicating that severe inflammation, exaggerated oxidative stress and myocardial contractile dysfunction, are the main contributors in promoting organ failure during sepsis progression [20,21,22]. The patients at a high risk of cardiac arrhythmias had increased intracellular Ca^2+^, which triggered activity via delayed afterdepolarizations and a shortening of the atrial refractory period. In sepsis, less is known of the effect exerted by Lemnalol in severe septic cardiac compilations. However, current therapies for this dangerous disease are not effective, and our understanding is nonspecific and requires further elucidation. 

## 2. Results

### 2.1. Effects of Lemnalol on the Action Potential of LA Myocytes 

To evaluate the effect of Lemnalol on the action potential of LA myocytes, we used LPS (1 μg/mL) to pretreat LA myocytes, and added Lemnalol (0.1 μM) to measure the difference in the action potential duration (APD) at 20% (APD_20_), 50% (APD_50_) and 90% (APD_90_) repolarization of the amplitude. Figure 1 shows the AP of the control, LPS-treated, and LPS + Lemnalol-treated LA myocytes. There was a significant difference in APD; however, there were similarities in the AP amplitude (APA) and resting membrane potential among the control, LPS, and LPS + Lemnalol groups. The LPS-treated LA myocytes had shorter APD_20_, APD_50_ and APD_90_ when compared with the controls. Moreover, APD_20_, APD_50_ and APD_90_ in the LPS + Lemnalol-treated LA myocytes were longer than those in the LPS-treated LA myocytes.

### 2.2. Effects of Lemnalol on the Membrane Currents of LA Myocytes

This section details the effect of Lemnalol on the Na^+^ current (*I*_Na_), late sodium current (*I*_Na-Late_), L-type Ca^2+^ current (*I*_Ca-L_), Na^+^/Ca^2+^ exchanger (NCX) current, transient outward current (*I*_to_) and the rapid component of the delayed rectifier potassium current (*I*_Kr-tail_). No significant difference was observed between the baseline values in *I*_Na_ among the control, LPS and LPS + Lemnalol groups (Figure 2A). Compared with the controls, the LPS-challenged LA myocytes exhibited a significantly larger *I*_Na-Late_ (*p* < 0.05) (Figure 2B). In addition, the *I*_Na-Late_ in the LPS + Lemnalol-treated LA myocytes was significantly lower than in the LPS-treated LA myocytes.

As shown in Figure 3, *I*_Ca-L_ in the LPS-treated LA myocytes was significantly lower than that in the controls, with a 34.33% decrease in the peak current (elicited from −50 to +10 mV). Compared with the LPS-treated LA myocytes, the LPS + Lemnalol-treated LA myocytes had a larger *I*_Ca-L_, demonstrating a 59.40% increase in the peak current (elicited from −50 to +10 mV). 

Both the modes of the NCX current were larger in the LPS-treated LA myocytes (Figure 4) than in the controls, with a 38.99% and 121.24% increase in the peak current in the forward and reverse modes (both elicited from −40 to +100 mV), respectively. Thus, the presence of Lemnalol reduced both the modes of the NCX current, with a 30.48% and 21.35% decrease in the peak current in the forward and reverse modes (both elicited from −40 to +100 mV), respectively. 

Compared with the controls, the *I*_to_ of the LPS-treated LA myocytes was larger, exhibiting a 52.4% increase in the peak current (elicited from −40 to +60 mV). Moreover, the presence of Lemnalol reduced the *I*_to_, displaying a 32.70% decrease in the peak current (elicited from −40 to +60 mV; Figure 5). The LPS-treated LA myocytes had a larger *I*_Kr-tail_ than the controls, with a 51.91% increase in the peak current (elicited from −40 to +60 mV), but the LPS + Lemnalol-treated LA myocytes displaying a 33.22% decrease in the peak current (Figure 6). The effect that LPS exerts on *I_Kr-tail_* of LA myocytes was almost completely abolished by adding Lemnalol.

### 2.3. Effects of Lemnalol on Calcium Handling of LA myocytes

As shown in Figure 7A, the LPS-treated LA myocytes exhibited a lesser amplitude of intracellular calcium ([Ca^2+^]_i_) transients than the controls. Following incubation with LPS + Lemnalol, the amplitude of the [Ca^2+^]_i_ transients increased further. 

### 2.4. Effects of Lemnalol on Protein Expression in LA Myocytes

LPS treatment was associated with significantly increased TNF-α and phosphorylated NF-κB P65 in rabbit LA myocytes (Figure 8). Adding Lemnalol reduced the LPS-induced production of TNF- α and the phosphorylation of NF-κB P65 (Figure 8). LPS impaired SR calcium uptake (Figure 9A). Lemnalol partially reversed LPS-induced sarcoendoplasmic reticulum calcium transport ATPase (SERCA2a) activity (Figure 9A). The administration of LPS profoundly enhanced the phosphorylation of RyR, but the expression of Cav1.2 only increased slightly (Figure 9C). Lemnalol considerably restored the LPS-induced phosphorylation of RyR (Figure 9B) and showed no statistically significant effect on the LPS-induced expression of Cav1.2 (Figure 9C).

## 3. Discussion 

The aim of the current study was to demonstrate the effect of Lemnalol in septic myocardial dysfunction, which may benefit the treatment of sepsis-induced arrhythmogenesis. Previous studies have shown no evidence that Lemnalol could be a potential target for antiarrhythmic therapy use [23]. The anti-arrhythmic effects of Lemnalol on LA indicate that it may play a role in LA arrhythmogenesis during the early stages of arrhythmogenesis, and the inhibition of calcium overload can reduce the persistence and genesis of arrhythmias. Cardiovascular disorders are the most common cause of death in most developed countries, including the United States of America [24]. AF is associated with a higher risk of thromboembolism, increased mortality and high healthcare expenditure in the recent decades [25,26,27,28,29,30], however, it is not life threatening. Ventricular fibrillation (VF) and ventricular tachycardia (VT) following AF are the most common causes of sudden cardiac death (SCD) [31].

Some studies have demonstrated that arrhythmogenic phenomena at the cardiomyocyte level are mainly characterized by a shortened AP without a plateau phase [32]. In the LPS-treated myocytes, APD was significantly shortened under sepsis, as previously reported [33]. A similar picture as this report was shown, and all of the APD_20,_ APD_50_ and APD_90_ of LA myocytes challenged with LPS were significantly shorter than those of the control myocytes (Figure 1) in this study. Lemnalol treatment on LA myocytes fully recovered the alterations caused by LPS.

Accumulating evidence indicates that *I*_Ca-L_ reduction could mediate arrhythmia in cardiomyocytes [34,35,36]. One of the major mechanisms of AP shortening during arrhythmia is the reduction of *I*_Ca-L_. The LPS-treated *I*_Ca-L_ was significantly reduced when compared with the control in LA myocytes, and the effect of that LPS exerts on *I*_Ca-L_ was almost completely reverted by Lemnalol (Figure 3). These results confirm that Lemnalol may regulate the Ca^2+^ influx via *I*_Ca-L_ on the sarcolemma reticulum (SR) and reverse APD alterations caused by LPS.

The enhanced Na^+^ influx will cause higher [Na^+^]_i_. Our data showed that pretreatment with Lemnalol had no effect on the *I*_Na_ current in the LPS-stimulated LA myocytes (Figure 2A). The enormous *I*_Na_ peak current is always tailed by one small late sodium current (*I*_Na-late_) during an AP event. The arrhythmogenic APs have a close relationship with the continuous flow of *I*_Na-Late_ during the plateau of AP, when enough intracellular sodium ions accumulate to affect APD [37]. We found that the *I*_Na-Late_ flux was profoundly increased by LPS, and it was abolished by co-treatment with Lemnalol in LA myocytes, in this study (Figure 2B). 

NCX is the most important contributor to the influx of Na^+^ and the main path for the exchange of Ca^2+^ with Na^+^. Thus, NCX works together with the increased *I*_Na-late_, causing calcium overload, and eventually, AF [38,39,40]. Arrhythmogenesis caused by calcium overload was induced by NCX in the SR. It may be reduced by decreasing [Na^+^]_i_ to revert the mode of NCX, in order to establish an electrochemical gradient with enough driving force to expel [Ca^2+^]_i_ from the sarcoplasmic reticulum (SR) [41,42,43,44]. The NCX flux enhanced by LPS can be significantly ameliorated by Lemnalol treatment (Figure 4). Lemnalol modulated the I_NCX_ and reduced the *I*_Na-Late_ under the influence caused by LPS, and acted as a potentially selective Na^+^ channel blocker for LA arrhythmogenesis.

However, the intracellular Ca^2+^ content was largely reduced (Figure 7) and the SR Ca^2+^ content was increased (Figure 7B) in LPS-challenged LA myocytes. Lemnalol abolished these LPS-induced changes on Ca^2+^ spark in LA myocytes (Figure 7). Both the forward and reverse modes of *I*_NCX_ operations were inhibited by Lemnalol under the influence of LPS treatment. Activity triggered by increased [Ca^2+^]_i_ may lead to arrhythmia, and this may be suppressed by decreasing *I*_Na-Late_, *I*_NCX_, and the other related potassium fluxes as well [38].

The potassium currents and *I*_Ca-L_ are the major components that work together to shape the repolarization picture of APs. Arrhythmogenesis caused by the shortening of AP was thought to contribute to *I*_K_ [33]. Thus, *I*_to_ and *I*_K_ modulation agents were thought to be potential anti-arrhythmia targets [45,46]. Our data revealed that *I*_to_ and *I*_K_ were profoundly enhanced under LPS stimulation in LA myocytes, and abolished by adding Lemnalol ( 5,6 ). The current study demonstrates that Lemnalol may suppress *I*_to_ and *I*_K_ to achieve anti-arrhythmic effects by slowing early repolarization to reduce the spontaneous firing rate.

Jean et al. (2008) reported that the *L. cervicorni* significantly inhibited the expression of COX-2 and iNOS in LPS-challenged RAW 264.7 cells [16]. In addition, Lemnalol was able to inhibit TNF-α mediated neuropathic pain in both the activated microglia and astrocytes in experimental animals [17]. Moreover, the study conducted by Lee et al. [18] further claimed that Lemnalol could attenuate nociceptive responses and chronic constriction injury-induced allydontia and thermal hyperplasia. Likewise, the results of a recent study conducted on *L. cervicorni* suggested that Lemnalol was able to produce significant inhibition of inflammation via inhibiting the activity of metalloproteinase 9 (MMP-9), tartrate- resistant acid phosphatase (TRAP) and cathepsin K expression [19].

Exaggerated inflammation may alter multiple ionic currents and cause reentry and/or focal electrical activity to induce arrhythmia. Furthermore, these overwhelmed inflammatory responses must not be overlooked during sepsis [47]. The binding of LPS to TLR4 enhances the translocation of NF-κB to nuclei (Figure 8), and this produces a pro-inflammatory cytokine shower in sepsis [48].

Cytokines, such as TNF-α (Figure 8) and IL-1β, are extensively produced at the early stage of sepsis [49]. It is well known that SERCA recycling of SR Ca^2+^ and RyRs release of SR Ca^2+^ during the diastolic phase fine tune Ca^2+^ [50]. Thus, the beneficial effects of Lemnalol on LPS-induced arrhythmogenesis may result from its anti-inflammatory effects.

## 4. Experimental Section

### 4.1. Materials

Lemnalol was provided by professor Chang-Yih Duh’s laboratory (National Sun Yat-Sen University, Kaohsiung, Taiwan).

### 4.2. Isolation of Single LA Myocytes

All the experiments conformed to the institutional *Guide for the Care and Use of Laboratory Animals*, and were approved by a local ethics review board (IACUC-19-085) and the Guide for the Care and Use of Laboratory Animals. Male rabbits, weighing 2 to 2.5 kg, were anesthetized intravenously and injected with sodium pentobarbital (100 mg/kg). The hearts were immediately removed and mounted on a Langendorff apparatus after anesthetization, in order to perform perfusion in 95% oxygenated normal Tyrode’s solution, at 37 °C, containing (mM): NaCl, 137; glucose, 11; HEPES, 10; KCl, 5.4; CaCl_2_, 1.8; MgCl_2_, 0.5. The pH was adjusted to 7.4 with NaOH. After the hearts were cleaned of all traces of blood, this perfusate was replaced with oxygenated Ca^2+^-free Tyrode’s solution, containing collagenase type I (300 units/mL) (Sigma Chemical, St. Louis, MO, USA) and protease type XIV (0.25 units/mL) (Sigma Chemical, St. Louis, MO, USA), for about 8–12 min, in order to perform enzymatic dispersion. The LA was then excised into small pieces and gently shaken in 50 mL of Ca^2+^-free oxygenated Tyrode’s solution, until single myocytes were isolated. The solution was then gradually changed to normal oxygenated Tyrode’s solution. The rabbit LA myocytes were allowed to stabilize in the bath for at least 30 min, before performing the experiments.

### 4.3. Electrophysiological Study

All of the whole-cell patch-clamp recordings were made from single LA myocytes in the control and in the presence of LPS (1 μg/mL), with and without Lemnalol (0.1 μM), using an Axopatch 1D amplifier (Axon Instruments, Sunnyvale, CA, USA) at 35 ± 1 °C. During this study, almost all of the experiments were unpaired, carried out on separate groups of cells: Under control conditions and the after incubation of cells for 6 ± 1 h in the presence of LPS, with and without Lemnalol. Thin-walled borosilicate capillary tubes (o.d. 1.8 mm) were used to fabricate microelectrodes with tip resistances of approximately 3–5 MΩ. 

Before the formation of a stable membrane-pipette seal, the tip potentials were zeroed in Tyrode’s solution. The junction potentials between the bath and pipette solution (9 mV) were also corrected for AP recordings. Action potentials were recorded in the current-clamp mode and ionic currents were measured in the voltage-clamp mode.

A small hyperpolarizing step from a holding potential of −50 mV to a testing potential of −55 mV for 80 ms, was conducted at the beginning of each experiment. The total cell capacitance was determined by the area obtained under the capacitive currents, which was divided by the applied voltage step. Normally, 60%–80% series resistance (Rs) was electronically compensated. The RMP was obtained during the period between the last repolarization and the onset of the subsequent AP. The APA was obtained from the measurement of RMP, to the peak of the AP depolarization. All the AP durations were measured at APD_20_, APD_50_ and APD_90_ repolarization of the amplitude stimulated with a rate of 1 Hz in all the preparations.

The filling solution used in the micropipettes consisted of (mM) CsCl, 130; HEPES, 10; MgATP, 5; Na_2_ phosphocreatine, 5; MgCl_2_, 1; NaGTP, 0.1 (adjusted to pH 7.2 with CsOH) for the experiments on the *I*_Ca-L_; with a solution consisting of (mM) CsCl, 133; TEACl, 20; EGTA, 10; HEPES, 5; MgATP, 5; NaCl, 5 (adjusted to pH 7.3 with CsOH) for the *I*_Na_. These consisted of (mM) CsCl, 130; Na_2_ATP, 4; MgCl_2,_ 1; EGTA, 10, and HEPES, 5 at pH 7.3 with NaOH for the late sodium current (*I*_Na-Late_); containing (in mM) CsCl, 110; NaCl, 20; MgCl_2_, 0.4; CaCl_2_, 1.75; TEACl, 20; HEPES, 10; BAPTA, 5; glucose, 5; MgATP, 5 (adjusted to pH 7.25 with CsOH) for the experiments in the NCX current. Those contained (mM) K aspartate, 110; KCl, 20; HEPES, 10; MgATP, 5; Na_2_ phosphocreatine, 5; MgCl_2_, 1; EGTA, 0.5; LiGTP, 0.1, and the pH was adjusted to 7.2 with KOH for the experiments in the AP and potassium currents.

The *I*_Na_ was recorded during depolarization, from a holding potential of −120 mV, to testing potentials ranging from −90 to +60 mV in 10 mV steps for 40 ms, at a frequency of 3 Hz and a room temperature of 25 ± 1 °C, unless otherwise specified. External solution containing (mM): CsCl, 133; NaCl, 5; HEPES, 5; glucose, 5; MgCl_2_, 2; CaCl_2_, 1.8; nifedipine, 0.002, with pH of 7.3, was used.

The *I*_Na-Late_ was recorded at room temperature with an external solution containing (in mM) NaCl, 130; CsCl, 5; MgCl_2,_ 1; CaCl_2,_ 1; HEPES, 10, and glucose 10 at pH 7.4 with NaOH, using a step/ramp protocol (−100 mV stepped to +20 mV for 100 ms, then ramped back to −100 mV over 100 ms). The *I*_Na-late_ was measured as the tetrodotoxin (30 µM)-sensitive portion(s) of the current traces obtained when the voltage was ramped back to −100 mV.

The *I*_Ca-L_ was measured as an inward current during depolarization from a holding potential of −50 mV to testing pulses ranging from −40 to +60 mV in 10 mV steps for 300 ms, at a frequency of 0.1 Hz, by means of a perforated patch clamp with amphotericin B. The NaCl and KCl in the normal Tyrode’s solution were replaced by TEACl and CsCl, respectively. In order to avoid the ‘run-down’ effects, *I*_Ca-L_ was measured on the peak inward current and the current at the end of each test pulse.

The NCX current was elicited by test potentials between −100 and +100 mV, from a holding potential of −40 mV for 300 ms, at a frequency of 0.1 Hz. The amplitudes of the NCX current were measured as 10 mM nickel-sensitive currents. The external solution (mM) contained NaCl, 140; glucose, 10; HEPES, 5; CaCl_2_, 2; MgCl_2_, 1; strophanthidin (10 μM); nitrendipine (10 μM), and niflumic acid (100 μM), and the pH was adjusted to 7.4.

The *I*_to_ was performed with a double-pulse protocol. A 30-ms pre-pulse ranging from −80 to −40 mV was used to inactivate the sodium channels, followed by a 300 ms test pulse to +60 mV in 10 mV steps at a frequency of 0.1 Hz. CdCl_2_ (200 μM), was added to the bath solution to inhibit *I*_Ca-L_. The I_to_ was measured as the difference between the peak outward current and the steady-state current.

The *I*_Kr-tail_ was measured as the outward peak tail current density following a three seconds pre-pulse from a holding potential of −40 mV to voltage between −40 and +60 mV in 10 mV steps at a frequency of 0.1 Hz in the presence of chromanol-293B (30 μM) and CdCl_2_ (200 μM) in the normal Tyrode’s solution. Micropipettes were filled with a solution containing (in mM) KCl, 120; MgCl_2_, 5; CaCl_2_, 0.36; EGTA, 5; HEPES, 5; glucose, 5; K_2_-ATP, 5; Na_2_CrP, 5; Na-GTP, 0.25 (adjusted to pH 7.2 with KOH).

### 4.4. Measurement of the Changes in the Intracellular Calcium and SR Calcium Contents

LA myocytes from the LPS-treated, LPS + Lemnalol-treated, and control groups were loaded with fluorescent Ca^2+^ (10 μM, Fluo-3 AM) for 30 min at room temperature, as described previously [8,24]. Fluo-3 fluorescence was excited by a the 488-nm line of an argon ion laser. The emission was recorded at λ > 515 nm. The cells were repetitively scanned at 2-ms intervals. Fluorescence imaging was performed with a laser scanning confocal microscope (Zeiss LSM 510, Carl Zeiss, Jena, Germany) and an inverted microscope (Axiovert 100, Carl Zeiss Jena, Germany). Fluorescent signals were corrected for the variations in dye concentrations by normalizing the fluorescence (F) against the baseline fluorescence (F_0_), to obtain reliable information regarding the transient intracellular Ca^2+^ [Ca^2+^]_i_ changes from the baseline values ((F - F0)/F0) and to exclude the variations in the fluorescence intensity from different volumes of the injected dye. The [Ca^2+^]_i_ transient, peak systolic [Ca^2+^]_i_ and diastolic [Ca^2+^]_i_ were measured during a 2-Hz field-stimulation with 10-ms twice-threshold strength square-wave pulses. 

### 4.5. Western Blot Analysis 

Control, LPS-treated, and LPS + Lemnalol-treated myocytes were centrifuged and washed with cold PBS, and lysed on ice for 30 min in RIPA buffer containing 50 mM Tris, pH 7.4, 150 mM NaCl, 1% NP40, 0.5% sodium deoxycholate, 0.1% sodium dodecylsulfate (SDS) and protease inhibitor cocktails (Sigma-Aldrich, St. Louis, MO, USA). The protein concentrations were determined with a Bio-Rad protein assay reagent (Bio-Rad, Hercules, CA, USA). The proteins were separated in 4%~12% SDS- polyacrylamide gel electrophoresis (PAGE) under reducing conditions and electrophoretically transferred into an equilibrated polyvinylidene difluoride membrane (Amersham Biosciences, Buckinghamshire, UK). All the blots were probed with primary antibodies against TNF-α, NF-κB, SERCA2a, phosphorylated RyR at Ser-2808 (RyR pS2808) (GeneTex, TX, USA) and α-actin (MBL, Nagoya, Japan), and all of the secondary antibodies were conjugated with horseradish peroxidase. All the bound antibodies were detected using an enhanced chemiluminescence detection system (Millipore, Billerica, MA, USA), and analyzed with the AlphaEaseFC software. All the targeted bands were normalized to α-actin to confirm equal protein loading.

### 4.6. Statistical Analysis

All the continuous variables are expressed as the mean ± S.E.M. The differences between the control, LPS and LPS + Lemnalol-treated LA myocytes, were compared using the Mann-Whitney rank sum unpaired t-test, depending on the outcome of the normality test. A *p* value of less than 0.05 was considered statistically significant.

## 5. Conclusion

LPS-induced events such as higher incidence of arrhythmogenesis and calcium homeostasis during sepsis status, and these results could be attenuate by Lemnalol suggest it may be a novel strategy of treatment for sepsis-induced cardia dysfunction.

## Figures and Tables

**Figure 1 marinedrugs-17-00619-f001:**
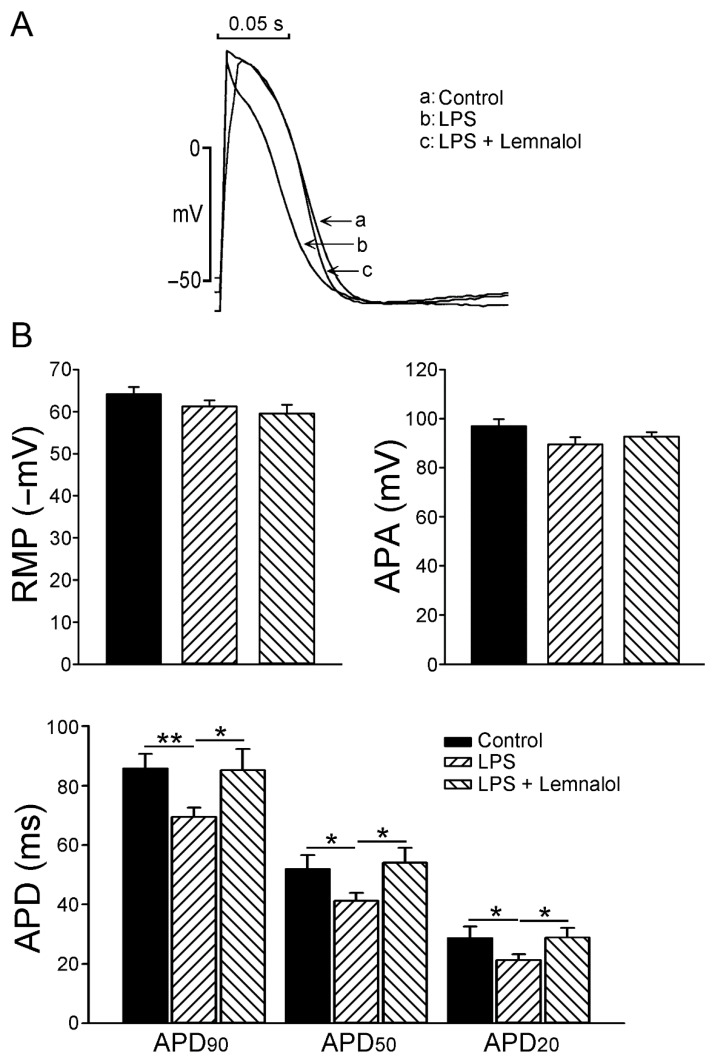
Action potentials (AP) of left atrial (LA) myocytes in control, lipopolysaccharide (LPS)-treated and LPS + lemnalol-treated LA myocytes. (**A**) Representative tracings and (**B**) average data of action potentials from control (*n* = 10), LPS-treated (*n* = 13) and LPS + lemnalol-treated (*n* = 10) LA myocytes. Resting membrane potential (RMP), action potential amplitude (APA), 20% of action potential duration (APD_20_), 50% of action potential duration (APD_50_), 90% of action potential duration (APD_90_). * *p* < 0.05; ** *p* < 0.01 compared to control group or LPS+lemnalol-treated group.

**Figure 2 marinedrugs-17-00619-f002:**
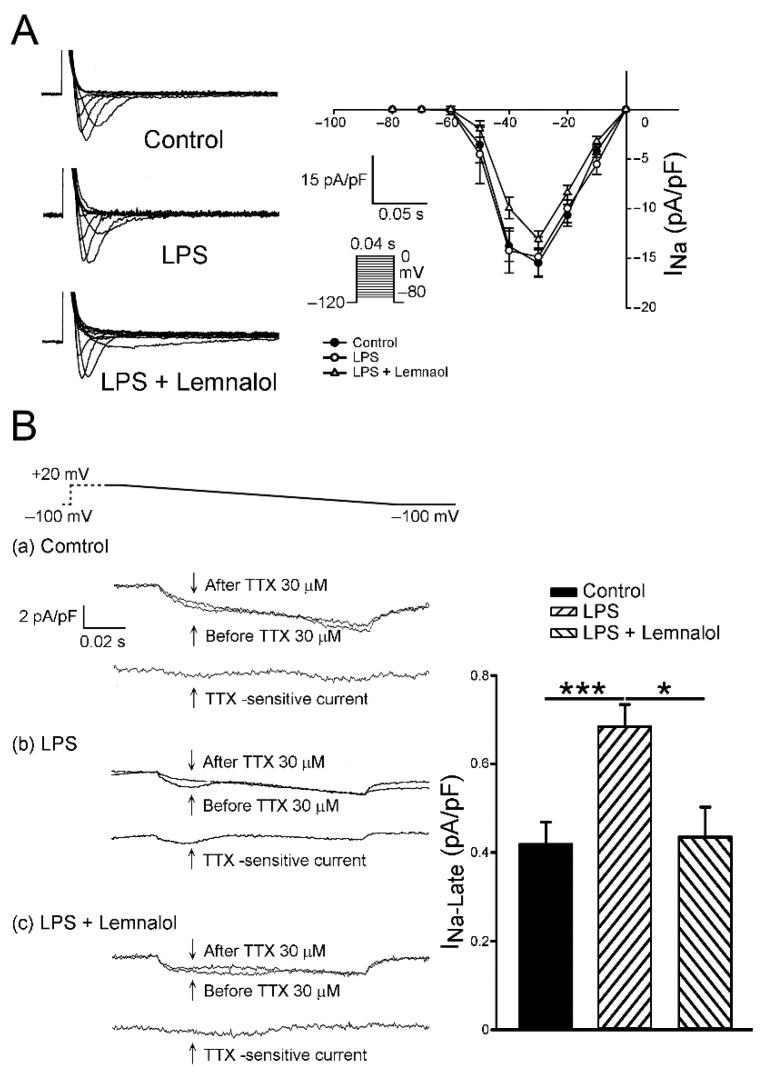
Sodium (*I*_Na_) and late sodium currents (*I*_Na-Late_) in control, LPS-treated and LPS + lemnalol-treated left atrial (LA) myocytes. (A) Representative current tracings and I–V relationship of the *I*_Na_ of LA myocytes from control (*n* = 12), LPS-treated (*n* = 14) and LPS + lemnalol-treated (*n* = 15) LA myocytes. (B) Representative current tracings and average data of the *I*_Na-Late_ of left atrial myocytes from control (*n* = 11), LPS-treated (*n* = 9) and LPS + lemnalol-treated (*n* = 9) LA myocytes. The insets in the current tracings show the various clamp protocols. * *p* < 0.05 compared to the LPS+lemnalol-treated group; *** *p* < 0.05 compared to the control group.

**Figure 3 marinedrugs-17-00619-f003:**
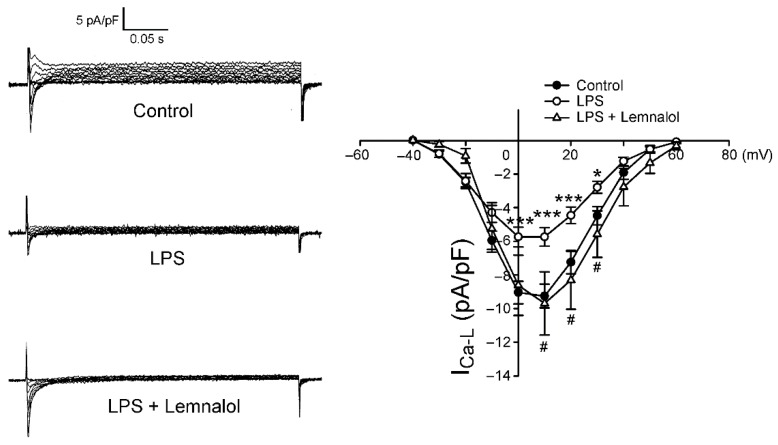
L-type calcium channel (*I*_Ca-L_) in control, LPS-treated and LPS + lemnalol-treated left atrial (LA) myocytes. The representative current tracings and I–V relationship of the *I*_Ca-L_ of LA myocytes from control (*n* = 15), LPS-treated (*n* = 15) and LPS + lemnalol-treated (*n* = 19) LA myocytes. * *p* < 0.05; *** *p* < 0.01 compared to control group; ^#^
*p* < 0.05 compared to LPS+lemnalol-treated group.

**Figure 4 marinedrugs-17-00619-f004:**
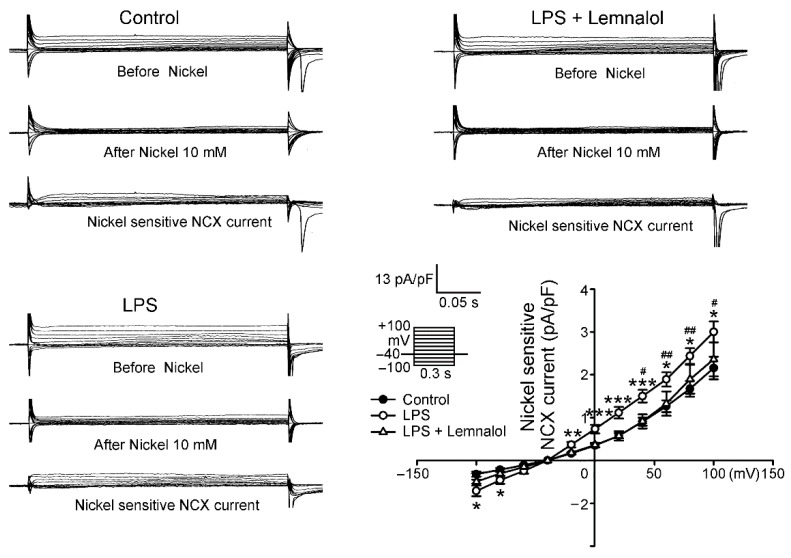
Na^+^–Ca^2+^ exchanger (NCX) current in control, LPS-treated and LPS + lemnalol-treated left atrial (LA) myocytes. Representative current tracings and I–V relationship of the NCX current of LA myocytes from control (*n* = 12), LPS-treated (*n* = 16) and LPS + lemnalol-treated (*n* = 9) LA myocytes. The insets in the current tracings show the various clamp protocols. * *p* < 0.05; ** *p* < 0.01; *** *p* < 0.005 compared to control group; ^#^
*p* < 0.05; ^##^
*p* < 0.01; ^###^
*p* < 0.005 compared to LPS+lemnalol-treated group.

**Figure 5 marinedrugs-17-00619-f005:**
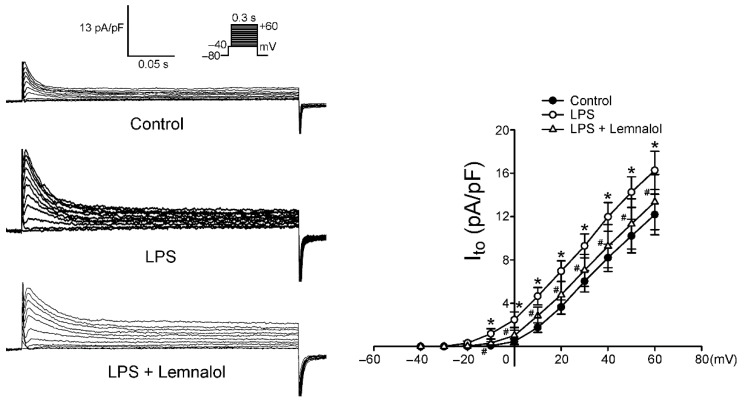
Transient outward current (*I*_to_) in control, LPS-treated and LPS + lemnalol-treated left atrial (LA) myocytes. Representative tracings and I–V relationship of the *I*_to_ of LA myocytes from control (*n* = 13), LPS-treated (*n* = 16) and LPS + lemnalol-treated (*n* = 15) LA myocytes. The inset in the top current tracings show the clamp protocol. ^#^
*p* < 0.05 compared to LPS+lemnalol-treated group; * *p* < 0.05 compared to control group.

**Figure 6 marinedrugs-17-00619-f006:**
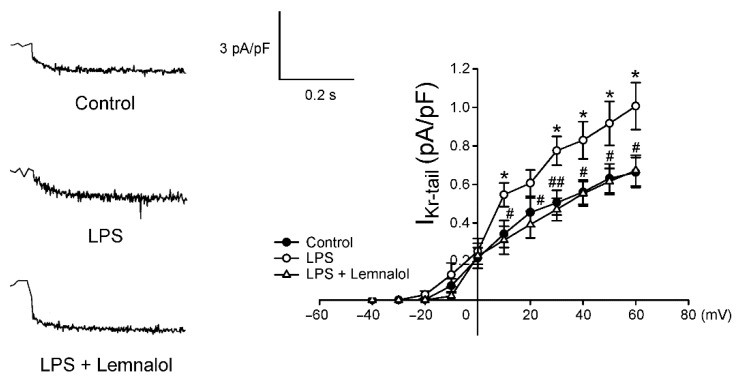
Delayed rectifier potassium current (*I*_Kr-tail_) in control, LPS-treated and LPS + lemnalol-treated left atrial (LA) myocytes. Representative tracings and I–V relationship of the *I*_Kr-tail_ of LA myocytes from control (*n* = 13), LPS-treated (*n* = 13) and LPS + lemnalol-treated (*n* = 10) LA myocytes. The insert in the representative current tracings shows the clamp protocol. * *p* < 0.05 compared to control group; ^#^
*p* < 0.05; ^##^
*p* < 0.05 compared to LPS+lemnalol-treated group.

**Figure 7 marinedrugs-17-00619-f007:**
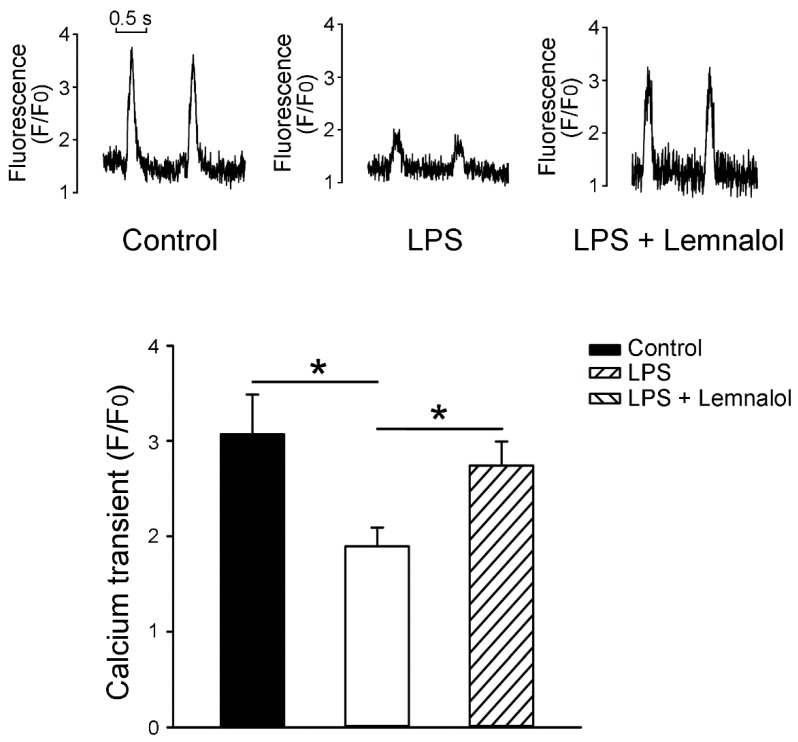
Intracellular Ca^2+^ ([Ca^2+^]_i_) transient measured from caffeine-induced Ca^2+^ transients in left atrial (LA) myocytes in control, LPS-treated and LPS + lemnalol-treated LA myocytes. Representative tracings and average data from the [Ca^2+^]_i_ transient in the LA myocytes recorded from control (*n* = 13), LPS-treated (*n* = 10) and LPS + lemnalol-treated (*n* = 11) LA myocytes; * *p* < 0.05 compared to control or LPS+lemnalol-treated group.

**Figure 8 marinedrugs-17-00619-f008:**
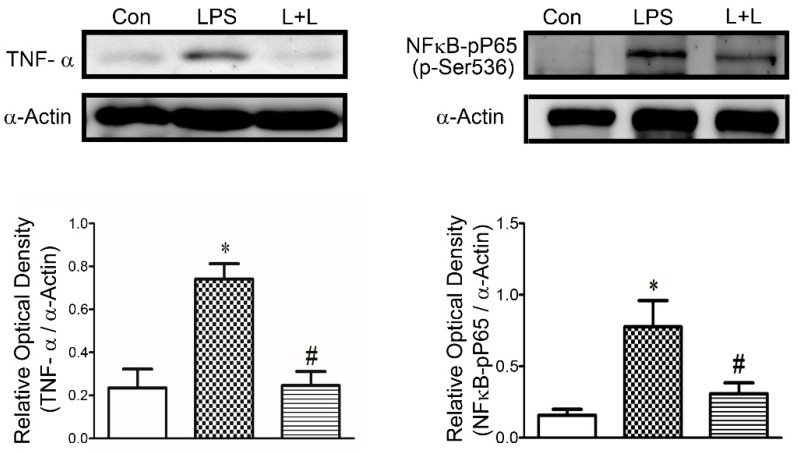
Inflammatory responses in control, LPS-treated and LPS + lemnalol-treated LA myocytes. Representative immunoblot and average data of TNF-α and phosphorylated NF-κB P65 at serine 536 (p-Ser536) from control, LPS-treated and LPS + lemnalol-treated LA myocytes. Data are expressed as mean ± SEM. Densitometry was normalized to α-actin as an internal control. * *p* < 0.05 compared to control group; ^#^
*p* < 0.05 compared to LPS+lemnalol-treated group.

**Figure 9 marinedrugs-17-00619-f009:**
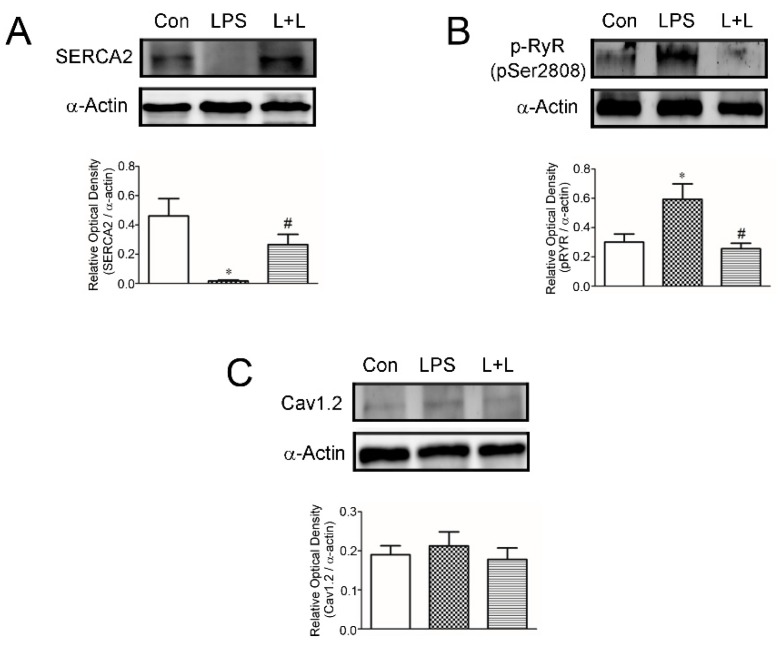
Ca^2+^ regulation proteins in control, LPS-treated and LPS + lemnalol-treated left atrial (LA) myocytes. Cell homogenates were analyzed by representative immunoblotting with sarcoendoplasmic reticulum calcium transport ATPase (SERCA2)-specific antibody (upper panel). Relative optical density of SERCA2 of myocyte homogenates were treated with LPS in the presence or absence of lemnalol (lower panel). (**A**) Representative immunoblot and average data of SERCA2 from control, LPS-treated and LPS + lemnalol-treated LA myocytes. SERCA2 levels were analyzed by immunoblotting of whole cell lysates. (**B**) Representative immunoblot and average data of p-RyR from control, LPS-treated and LPS + lemnalol-treated LA myocytes. The phosphorylated RyR were immunoblotted for Ser-2808 epitope (p-Ser2808). (**C**) The expression of Cav 1.2 did not show statistically significant increasing under LPS or LPS + lemnalol administration. Data are expressed as mean ± SEM. Densitometry was normalized to α-actin as an internal control. * *p* < 0.05 compared to control group; ^#^
*p* < 0.05 compared to LPS +lemnalol-treated group.

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
