# Peer review of "Lemnalol Modulates the Electrophysiological Characteristics and Calcium Homeostasis of Atrial Myocytes"

_marinedrugs, 2019, doi:10.3390/md17110619_

Round 1

Reviewer 1 Report

The manuscript is well done. The authors should improve the usability of Figure 1. They have to indicate with A and B upper and lower parts of it.

The authors should significantly improve the conclusions.

Minor concern: 

Minor concern:

Please change the title of paragraph 2.1. because for morphology the reader may suppose a histological image!

P.S. I think that the authors may improve highly the manuscript with a true morphological study!

Author Response

Responses to Reviewer #1

Thank you very much for your detailed comments. Those comments were very instructive, and very helpful to this manuscript.

Regard to the general comments: “. Please change the title of paragraph 2.1. because for morphology the reader may suppose a histological image!”

--We are sorry for the mistake on expression problems. We had deleted the “morphological” word to avoid readers cause misunderstanding.

Regard to the general comments: “The authors should significantly improve the conclusions.”

--We are very much thankful to you for your deep and thorough review, and are grateful for your positive comments.

According to your suggestion, we have improve the conclusion as follows “LPS-induced events such as higher incidence of arrhythmogenesis and calcium homeostasis during sepsis status, and these results could be attenuate by Lemnalol suggest it may be a novel strategy of treatment for sepsis induced cardia dysfunction.“ We have indicated the changes in red font in the revised manuscript.

The above descriptions are the responses to your comments and suggestions.

Sincerely yours,

Chih-Hsueng Hsu, MD, Ph. D

Reviewer 2 Report

Review comment sheet

1 A brief summary
   The paper describes that this study examined whether Lemnalol could modulate the electrophysiological characteristics and calcium homeostasis of atrial myocytes under the influence of LPS. So the conclusion is that Lemnalol modulates LPS-induced LA electrophysiological characteristics, calcium homeostasis, and blocks the inflammation mediated by NF-κB, which may contribute to the attenuating of LPS-induced arrhythmogenesis.    This is a very nice paper. However, I have some comments.

2 Overall evaluation
   The findings from this paper are excellent and deserves peer review of this journal.

3 Main problem

   This manuscript contained some questions described below.
   I think this paper has very interesting, this study contributes to future's clinical medicine largely. I have some questions from a point of view of clinical medicine.

   In the early phase of septic shock, there is a symptom that “limbs become warm and blood pressure decreases”. Needless to say, this is called Warm Shock. These symptoms occur due to the production of vasodilatory substances such as nitric oxide and increased cardiac output. Later, if sepsis worsens further, a condition called Cold Shock occurs. In the cold shock state, the blood vessels contract, the limbs get cold, the cardiac output decreases, and circulatory failure occurs. Sepsis can cause a variety of pathophysiological condition, but from this study, for example, what kind of electrophysiological changes can Lemnalol cause for cardiac ventricular muscles or peripheral blood vessels?

   And I want to know that what kind of pathological ( morphological ) changes do this experimental LA myocytes in this study ?

   Concerning to NF-κB, why did the author choose NF-κB ? What changes could be caused by other inflammatory cytokines, such as IL6, TNF-α or IL8 ?

   In humans, how much effect can be seen with the amount of this Lemnalol used in this study ? Please let me know that.

Author Response

Response to Reviewer #2

Thank you very much for your detailed comments. Those comments were very instructive, and very helpful to this manuscript. The responses to those comments are dictated below.

Regarding the specific comment “A brief summary

  -The paper describes that this study examined whether Lemnalol could modulate the electrophysiological characteristics and calcium homeostasis of atrial myocytes under the influence of LPS. So the conclusion is that Lemnalol modulates LPS-induced LA electrophysiological characteristics, calcium homeostasis, and blocks the inflammation mediated by NF-κB, which may contribute to the attenuating of LPS-induced arrhythmogenesis.    This is a very nice paper. However, I have some comments.”

Thank you very much for your detailed comments. Those comments were very constructive and helpful to this manuscript. Your suggestion deeply inspired our team.

Regarding the specific comment “ Overall evaluation

   The findings from this paper are excellent and deserves peer review of this journal.“

-Thank you very much for your comments. We thank you for your great recommendation.

Regarding the specific comment “In the early phase of septic shock, there is a symptom that “limbs become warm and blood pressure decreases”. Needless to say, this is called Warm Shock. These symptoms occur due to the production of vasodilatory substances such as nitric oxide and increased cardiac output. Later, if sepsis worsens further, a condition called Cold Shock occurs. In the cold shock state, the blood vessels contract, the limbs get cold, the cardiac output decreases, and circulatory failure occurs. Sepsis can cause a variety of pathophysiological condition, but from this study, for example, what kind of electrophysiological changes can Lemnalol cause for cardiac ventricular muscles or peripheral blood vessels?”

- We are very much thankful to you for your deep and thorough review, and are grateful for your positive and encouraging comments. I very much agree with your point of view about various pathophysiological conditions in different stages and organs. In the present study, we focus on left atrial (LA) myocytes LA myocytes. Because, many studies intense indicated that LA myocytes play an important role in development of atrial fibrillation (AF) (Wolf et al, 1991; Furberg et al, 1994; Psaty et al. 1997). And our previous study also found another anti-inflammatory property of coral-derived compound, excavatolide B also attenuated LPS-induced in alteration of calcium in LA myocytes. It may be very interesting of the effect of anti-inflammatory properties of coral-derived compound on LPS-challenged cardiac ventricular muscles or other tissues. In our preliminary observations, alternation of   (ICa-L, INa-Late, INCX, IKr-tail) after LSP challenge. The LPS-induced electrophysiological changes are different between LA myocyte and ventricular myocyte. It is worthy to investigate in future research works.

Reference:

Wolf P. A., Abbott R. D., Kannel W. B. “Atrial fibrillation as an independent risk factor for stroke: the Framingham Study,” Stroke 1991, 22, 8, 983–988. Furberg C. D., Psaty B. M., Manolio T. A., Gardin J. M., Smith V. E., Rautaharju P. M. Prevalence of atrial fibrillation in elderly subjects (the Cardiovascular Health Study), Am J Cardio 1994, 74, 3, 236–241. Psaty B. M., Manolio T. A., Kuller L. H., Incidence of and risk factors for atrial fibrillation in older adults, Circ 1997, 96, 7, 2455–2461.

Regarding the specific comment “And I want to know that what kind of pathological (morphological) changes do this experimental LA myocytes in this study?”

- Thanks for your careful comment. The present study did not have any the histopathological examinations. We had deleted the “morphological” word to avoid readers cause misunderstanding.

” Regarding the specific comment “Concerning to NF-κB, why did the author choose NF-κB? What changes could be caused by other inflammatory cytokines, such as IL6, TNF-α or IL8?

- We appreciate this comment very much and thanks for your comment. We actually did both NF-κB and TNF-α experiment in this study; please refer to Figure 8 in this manuscript for detail description. Since tumor necrosis factor act as a pro-inflammatory cytokine plays a vital role in sepsis-related cardiac function. And the SERCA2a has as important role in regulatory cardiac functions by Ca2+ homeostasis. TNF-α enhance the methylation of the SERCA2a promotor and this result in reducing SERCA2a (Kao et al. 2010). The current density of ICa-L was smaller but Ito, INCX and Iti were larger in the TNFα-treated cardiomyocytes versus control PV cardiomyocytes. Moreover, the sarcoplasmic reticulum Ca2+ content in the TNFα-treated cardiomyocytes was smaller than that in the control PV cardiomyocytes (Lee et al. 2007). These reports showed the same findings as we did in the study and correlated with the pathway from TNF-α to NF-κB under the influence of LPS.

Reference:

Kao Y.H., Chen Y. C., Cheng C. C., Lee T. I., Chen Y. J., Chen S. A. Tumor necrosis factor-α decreases sarcoplasmic reticulum Ca2-ATPase expressions via the promoter methylation in cardiomyocytes. Crit Care Med 2010, 38, 1, 217-222. Lee S. H., Chen Y. C., Chen Y. J., Chang S. L., Tai C. T., Wongcharoen W. et al. Tumor necrosis factor-α alters calcium handling and increases sarcoplasmic arrhythmogenesis of pulmonary vein cardiomyocytes. Life Sci 2007, 80, 1806-1015.

Regarding the specific comment “In humans, how much effect can be seen with the amount of this Lemnalol used in this study? Please let me know that.”

 - We appreciate and thanks for your comment very much. Our previous studies had found that systemic lemnalol (15mg/kg) and (30mg/kg) can significantly attenuate inflammatory pain and gouty arthritis in rats, respectively (Jean et al., 2008; Lee et al., 2013). The ratio between human and rat used to estimate the effective dosages of etoricoxib in rats according to the equivalent surface area dosage conversion factors was 1:7 (Freireich et al., 1966). Therefore, in my opinion, according to the conversion formula, the effective dosage of lemnalol for human is 2.15-4.28mg/kg.

Reference:

Jean Y. H., Chen W. F., Duh C. Y., Huang S.Y., Hsu C. H., Lin C. S., et al. Inducible nitric oxide synthase and cyclooxygenase-2 participate in anti-inflammatory and analgesic effects of the natural marine compound lemnalol from Formosan soft coral Lemnalia cervicorni. Euro J Pharmacol 2008, 578, 323-331.

Lee H. P., Huang S.Y., Lin Y. Y., Wang H. M., Jean Y. H., Wu S. F., et al. Soft coral-derived lemnalol alleviates monosodium urate-induces gouty arthritis in rats by inhibiting leukocyte infiltration and iNOS, COX-2 and c-Fos protein expression. Mar Drugs, 2013, 11, 99-113.

Freireich, E.J., Gehan, E.A., Rall, D.P., Schmidt, L.H., Skipper, H.E. Quantitative comparison of toxicity of anticancer agents in mouse, rat, hamster, dog, monkey, and man. CancerTreat Rep, 1966, 50, 4, 219-244.

The above descriptions are the responses to your comments and suggestions.

Sincerely yours,

Chih-Hsueng Hsu, MD, Ph. D

Round 2

Reviewer 2 Report

I think that the author has answered my question and comment appropriately and that this answer is properly reflected in the text.